# Prognostic Factors of Long-Term Outcomes after Primary Chemo-Radiotherapy in Non-Metastatic Anal Squamous Cell Carcinoma: An International Bicentric Cohort

**DOI:** 10.3390/biomedicines11030791

**Published:** 2023-03-06

**Authors:** Soledad Iseas, Diego Prost, Sarah Bouchereau, Mariano Golubicki, Juan Robbio, Ana Oviedo, Mariana Coraglio, Mirta Kujaruk, Guillermo Méndez, Marcela Carballido, Enrique Roca, Louis Gros, Vincent De Parades, Nabil Baba-Hamed, Julien Adam, Martín Carlos Abba, Eric Raymond

**Affiliations:** 1Oncology Unit, Gastroenterology Hospital “Dr. Carlos Bonorino Udaondo”, Av. Caseros 2061, Buenos Aires C1264, Argentina; 2Medical Oncology Department, Paris-St Joseph Hospital, 185 Rue Raymond Losserand, 75014 Paris, France; 3INSERM CNRS, UMRS 1127, ICM, QP-HP, Hôpitaux Universitaire La Pitie Salpêtrerie, Sorbonne Université, 75006 Paris, France; 4Pathology Unit, Paris-St Joseph Hospital, 185 Rue Raymond Losserand, 75014 Paris, France; 5Proctology Unit, Gastroenterology Hospital “Dr. Carlos Bonorino Udaondo”, Av. Caseros 2061, Buenos Aires C1264, Argentina; 6Pathology Unit, Gastroenterology Hospital “Dr. Carlos Bonorino Udaondo”, Av. Caseros 2061, Buenos Aires C1264, Argentina; 7Proctology Unit, Paris-St Joseph Hospital, 185 Rue Raymond Losserand, 75014 Paris, France; 8Basic and Applied Immunological Research Center (CINIBA), School of Medical Sciences, National University of La Plata, Calle 60 y 120, La Plata C1900, Argentina

**Keywords:** ASCC, prognostic factors, outcomes, multicenter cohort

## Abstract

Anal squamous cell carcinoma (ASCC) is a rare malignancy with a rising incidence associated with human papillomavirus (HPV) infection. The locally advanced disease is associated with a 30% rate of treatment failure after standard chemoradiotherapy (CRT). We aimed to elucidate the prognostic factors for ASCC after curative CRT. A retrospective multicenter study of 176 consecutive patients with ASCC having completed CRT treated between 2010 and 2017 at two centers was performed. Complete response (CR), disease-free survival (DFS), and overall survival (OS) were analyzed by Kaplan–Meier estimates with log-rank tests. The hierarchical clustering on principal components (HCPC) method was employed in an unsupervised and multivariate approach. The CR rate was 70% and was predictive of DFS (*p* < 0.0001) and OS (*p* < 0.0001), where non-CR cases were associated with shorter DFS (HR = 16.5, 95% CI 8.19–33.21) and OS (HR = 8.42, 95% CI 3.77–18.81) in a univariate analysis. The median follow-up was 38 months, with a 3-year DFS of 71%. The prognostic factors for DFS were cT1-T2 (*p* = 0.0002), N0 (*p* = 0.035), HIV-positive (*p* = 0.047), HIV-HPV coinfection (*p* = 0.018), and well-differentiated tumors (*p* = 0.037). The three-year OS was 81.6%. Female sex (*p* = 0.05), cT1-T2 (*p* = 0.02) and well-differentiated tumors (*p* = 0.003) were associated with better OS. The unsupervised analysis demonstrated a clear segregation of patients in three clusters, identifying that poor prognosis clusters associated with shorter DFS (HR = 1.74 95% CI = 1.25–2.42, *p* = 0.0008) were enriched with the locally advanced disease, anal canal location, HIV-HPV coinfection, and non-CR. In conclusion, our results reinforce the prognostic value of T stage, N stage, sex, differentiation status, tumor location, and HIV-HPV coinfection in ASCC after CRT.

## 1. Introduction

Anal squamous cell carcinoma (ASCC) stands as a rare disease, accounting for <3% of all gastrointestinal neoplasms [1]. The incidence of ASCC has been steadily increasing by 2.2% per year for the last three decades, notably in high-income countries [2]. Approximately 50,000 new cases and 19,000 deaths were anticipated worldwide in 2020 [2,3]. ASCC is more common in women than men and patients are typically diagnosed in their 60s [4]. Human papillomavirus (HPV) infection, human immunodeficiency virus (HIV), infection-related immune depression, immunosuppressive drugs for transplantation, autoimmune diseases, as well as smoking have been identified as major risk factors for ASCC [5,6]. HPV infection is found in nearly 90% of ASCC, mostly HPV-16, followed by other high-risk genotypes (e.g., HPV-18, -31, -33, -45) [7]. Compared to the general population, those with HIV have a 30-fold increased chance of developing ASCC [6,8]. The elevated incidence of HPV-related cancers in this population may be brought on by an interplay between latent HPV infection and immune suppression brought on by HIV [4].

The key factor that influences the ASCC outcome remains the stage at diagnosis [9]. Approximately 85% of patients have localized disease at diagnosis [10]. The primary aim of curative treatment is to achieve locoregional control while preserving anal function, avoiding a colostomy and maintaining a reasonable quality of life [4,11,12]. CRT with mitomycin C combined with 5-fluorouracil (5-FU) has been established as the standard of care since early 1990. Several randomized phase three trials have demonstrated the benefit of combining 5-FU and mitomycin C with radiotherapy compared to radiotherapy alone, and this concurrent combination was demonstrated to be superior to radiotherapy with 5-FU alone for both the local and locoregional disease [13,14]. Thus, for cT1-T2 tumors, complete regression is achieved in 80–90% of patients, with local recurrence rates of 15% and a 5-year survival of about 85% for a tumor size of less than centimeters [15]. In contrast, cT3-T4 and/or node-positive ASCC yields long-term recurrence rates of 20–44% [16]. For instance, the 5-year disease-free survival (DFS) rate was 47% in tumors greater than five centimeters, and 35% in cases of nodal involvement [12,16]. Several intensive approaches have been developed in patients with locally advanced disease, including high-dose radiotherapy [17] and a combination with anti-EGFR antibodies [18]. Neoadjuvant/adjuvant chemotherapy regimens have also been evaluated but failed to improve long-term outcomes in ASCC [19]. Salvage abdominal perineal resection results in 5-year locoregional control in 30–77% of patients [20]. Metastatic recurrences are associated with a poor prognosis with a median overall survival (OS) ranging from 12 to 20 months [21]. Based on data in other epidermoid carcinomas, checkpoint inhibitor immunotherapy combined with taxane-based chemotherapy is currently being evaluated in combination with radiotherapy in the non-metastatic setting in low-risk ASCC patients [12].

Given the rarity of ASCC, a comprehensive characterization of its molecular landscape is lacking. The few available studies indicate that ASCC presents a specific immune microenvironment, and genomic and transcriptomic abnormalities that may help identify biological parameters of interest for prognosis and targeted therapies [22,23]. We performed a retrospective multicenter study aimed at elucidating the clinical and pathological factors impacting local and distant DFS for non-metastatic ASCC (NM-ASCC) patients who have undergone curative CRT.

## 2. Materials and Methods

### 2.1. Anal Cancer Cohort

This multicenter cohort and comparative retrospective study comprised 176 consecutive eligible non-metastatic anal cancer patients recruited between 2010 and 2017 who were treated with curative intent at the oncology units of Hopital Paris Saint Joseph (HPSJ) in Paris, France, or Hospital Bonorino Udaondo (HBU) in Buenos Aires, Argentina. The protocol was approved by the ethics committees of both institutions. Patients had to provide informed consent for their data collection according to the recommendation of the ethics committee and in accordance with the European Union GDPR before participating in the study.

To be eligible, patients had to be at least 18-years-old, have an available pretreatment formalin-fixed paraffin-embedded (FFPE) biopsy, histologically confirmed squamous cell carcinoma, TNM clinical stage I–III disease, and have completed definitive CRT as their primary therapeutic approach. Patients with anal adenocarcinoma or in situ squamous cell carcinoma were not eligible. Clinical and pathologic data were retrieved from patient electronic and/or paper records.

Initial clinical staging was based on an anoscopy and digital anal examination, thorax–abdomen computed tomography (CT) scan, pelvic magnetic resonance imaging (MRI), and bloodwork (complete blood count, electrolytes, liver and renal function tests, HIV test). FDG-PET/TC was performed at baseline at HPSJ only. Data collected from patient medical records included age at diagnosis, sex, tumor location, HIV status, HPV status, cTNM, treatment response assessment, salvage surgery, recurrence data, relapse patterns, and survival status.

### 2.2. Treatment and Follow-Up

Radiotherapy was performed using either 3D conformal or intensity-modulated techniques (IMRT), with a median dose of 54 Gy in 30 daily fractions over 5.5 weeks. The three chemotherapy regimens delivered concomitantly with radiotherapy were: (1) mitomycin C 12 mg/m^2^ intravenous (IV) bolus, day 1–29 (maximum dose 20 mg) with 5-FU 1000 mg/m^2^ on days 1–4 and 29–32 by continuous 24-h IV infusion; (2) mitomycin C 12 mg/m^2^ IV bolus, day 1 only (maximum dose 20 mg) with capecitabine 825 mg/m^2^ twice daily on each radiotherapy treatment day; and (3) cisplatin 60 mg/m^2^ on days 1 and 29, with 5-FU 1000 mg/m^2^ on days 1–4 and 29–32 by continuous 24-h IV infusion. All HIV-positive patients received concomitant highly active antiretroviral therapy (HAART). Patients with bulky tumors who were highly symptomatic at presentation received induction chemotherapy before CRT. All cases were discussed by a multidisciplinary team in each respective center. The choice of chemotherapy regimen was per the physician’s discretion.

After completing the treatment, the response was determined according to Response Evaluation Criteria in Solid Tumors (RECIST) 1.1 from clinical, anorectoscopy, and radiological images at 24 weeks. The complete response (CR) was defined as clinical (on anal inspection and examination), radiological (CT and MRI of the pelvis), and rectoscopy disappearance of the disease (no evidence of disease; suspected lesions were confirmed by a biopsy). Abdominal perineal resection was performed in non-CR patients. Clinical follow-ups included digital anal examination, anoscopy, and a clinical exam, monthly for 3 months, then every 3 months for the first 2 years, then every 6 months for 3 to 5 years. Thorax CT and abdominal–pelvic MRI imaging were performed every 6 months during the follow-up period.

### 2.3. HPV Detection and Genotyping

FFPE tissues were reviewed by a specialist pathologist to confirm the presence of invasive ASCC. For the HPSJ cohort, slides were cut from a representative block from each tumor and stained for the p16 surrogate marker for HPV using the automated DAKO autotimer. Staining for p16 was carried out using the mouse antihuman monoclonal p16 antibody. Slides were categorized as p16-positive or negative by a pathologist blinded to clinical outcomes. The p16 was considered absent if <5 cells stained positive. For the HBU cohort, HPV detection was performed using purified genomic DNA and polymerase chain reaction (PCR) with biotinylated Broad-Spectrum General Primers BSGP5+/GP6+ designed to amplify a 140 bp fragment of the HPV-L1 gene. Genotyping was carried out by reverse line blot hybridization which identifies 36 HPV genotypes (Artisan technique, validated by WHO HPV LabNet) [24,25]. Briefly, the denatured biotinylated amplicons were hybridized with genotype-specific oligonucleotide probes immobilized as parallel lines on membrane strips.

### 2.4. Statistical Analysis

The Shapiro–Wilk test was employed for the assessment of normality. To compare categorical data between groups, the Chi-square test was used. Continuous data were compared with the Wilcoxon rank-sum test. Survival curves were estimated using the Kaplan–Meier method and compared using the log-rank test. Endpoints were the CR rate, DFS, and OS. The DFS was defined as the time from the first day of CRT until clinical or radiological disease recurrence or death from any cause. OS was defined as the time from treatment initiation to death from any cause. Cox proportional-hazards models were used to determine the association between OS or DFS and predictive variables, and were expressed as hazard ratios (HR) with their corresponding 95% confidence intervals (CI). Two-tailed *p*-values were calculated and the *p*-values < 0.05 were considered significant.

The Hierarchical Clustering on Principal Components (HCPC) method provided by the FactoMineR R package [26] was employed to identify patient clusters in an unsupervised and multivariate approach. The clinicopathological variables included were sex, HIV status, differentiation status, disease location, HIV-HPV coinfection, cT1-T2, cT3-T4, CRT treatment response, and disease progression (local recurrence, metastasis, or death). Briefly, the Principal Component method was used as a preprocessing step for the clustering to denoise the data and to balance groups of variables included in the model. The Principal Component analysis representation was also used to visualize the hierarchical tree and/or the partitions before the hierarchical clustering of patients based on Euclidean distances. Cluster characterization was performed by visual representation of the v-test values associated with variables that were significantly contributing to the clusters partition (*p* < 0.05). All statistical data analyses were performed using R Statistical Software, version 4.2.0 (Vienna, Austria).

## 3. Results

### 3.1. Patients

A total of 176 consecutive NM-ASCC patients with available primary FFPE tissue were included. The demographics, clinical presentation, treatment, and outcomes are summarized in Table 1. The median age was 61 years (range 26–89). When grouped by age at initial diagnosis, 11% (*n* = 20/176) were younger than 50 years, 31% (*n* = 55/176) were between 50 and 60 years and 58% (*n* = 101/176) were older than 60 years. Female patients were more frequently affected than males across all age groups (68%) and the male to female ratio was 1:2. Approximately half of the population (54%) had stage III disease and ASCC tumors were moderately differentiated in half of the cases. HPV and HIV-positive tumors represented 93% and 22% of the overall cohort, respectively. Overall, the baseline characteristics were well-balanced between two cohorts by center. Of note, the HPSJ cohort was significantly enriched in HPV-positive tumors and had a higher trend in the node involvement rate at the baseline clinical staging, possibly because of the immunosuppression induced by HIV. PET-CT vs. MRI disagreement was observed in 30/97 cases. In these patients, PET-CT changed Stage II to III in 9 cases and added more nodal burden disease for the remaining 21 patients.

Of note, no significant difference was found in the CR rate, DFS, and OS between the HPSJ and HBU cohorts (*p* = 0.68, *p* = 0.72, *p* = 0.84, respectively). Below we describe the potential prognostic value of the baseline demographic and clinicopathological factors, and viral infections for the overall cohort, in relation to oncological outcomes.

### 3.2. Treatment and Response

Overall, 114 patients (65%) received mitomycin C/5-FU (mitomycin C 1 dose: 20%, mitomycin C 2 doses: 80%), 40 patients received mitomycin C/capecitabine (23%), and 21 patients received cisplatin/5-FU (12%) concomitantly with 3D/IMRT pelvic radiotherapy. It is worth noting that the regimen of choice in the French cohort was based on mitomycin C + 5-FU, while in the Argentinian cohort, different validated concurrent schemes were used. In the HPSJ cohort, 42 patients received induction chemotherapy before CRT per physician discretion based on bulky disease, symptoms, and comorbidities, including 30 patients with therapy based on cisplatin/5-FU (71%), seven patients with carboplatin/5-FU (17%), three patients with docetaxel/cisplatin/5-FU (7%), and two patients with FOLFOX (5%).

The CR rate at 24 weeks was 70% in the overall cohort (Table 1). None of the following clinical variables evaluated—age, sex, differentiation status, HPV or HIV status, or HIV-HPV coinfection—showed prognostic value with respect to CR rates (*p* > 0.05). On the other hand, cT1-T2 vs. cT3-T4, and negative nodal involvement were significantly associated with higher CR rates (*p* = 0.05 and *p* = 0.01, respectively) (Appendix A).

### 3.3. Outcomes during Follow-Up

With a median follow-up after treatment completion of 38 months (range 6–149 months), 51 patients (29%) had disease progression. Thirty-seven patients relapsed at the primary site only, nine patients presented with distant metastases only at recurrence, and five patients had evidence of local and metastatic recurrence. Lost follow-up and missing data were observed only in 1 patient. There was no statistically significant difference between two cohorts in terms of DFS (*p* = 0.72) (Figure 1). The 3-year DFS rate was 71%. CR to CRT was a reliable surrogate predictive factor of DFS (*p* < 0.0001) and OS (*p* < 0.0001), where non-CR cases were associated with shorter DFS (HR = 16.5, 95% CI 8.19–33.21) and OS (HR = 8.42, 95% CI 3.77–18.81) in univariate analysis. In addition, the cT1-T2 (*p* = 0.0002), absence of nodal involvement (*p* = 0.035), HIV-positive (*p* = 0.047), HIV-HPV coinfection (*p* = 0.018), well-differentiated tumors (*p* = 0.037), and standard CRT (*p* = 0.001) were also prognostic factors of DFS (Table 2). Sex and HPV status did not significantly predict DFS (*p* > 0.05). At recurrence, we did not observe differences in DFS for local, distant, and synchronous relapse patterns (*p* > 0.05).

For the whole cohort, the OS rate at 3 years was 81.6%. There was no statically significant difference in OS between both cohorts (*p* = 0.84) (Appendix A). The univariable analysis of OS showed that female sex (*p* = 0.05), cT1-T2 (*p* = 0.02), and well differentiated tumors (*p* = 0.003) were associated with better OS (Appendix A). In contrast, HIV, HPV and HIV-HPV coinfection, nodal disease, and tumor location were not significantly associated with OS.

### 3.4. Multivariate Analysis of NM-ASCC Patients

The HCPC approach was used to find groups of NM-ASCC patients who shared clinical and pathological characteristics. Based on the similarity distances from dimensions 1 and 2 in the multidimensional scaling plot, the unsupervised analysis showed a clear segregation of NM-ASCC patients into three separate clusters according to 11 integrated variables (sex, age, T stage, N stage, differentiation, location, CRT, treatment response, HIV, HPV, and HIV-HPV coinfections) and excluding patient follow-up data (Figure 2a).

In contrast to Cluster 1, which was linked to a better prognosis, the univariate survival analysis showed that the NM-ASCC Clusters 2 and 3 were significantly related with a shorter DFS (HR = 1.74, 95% CI = 1.25–2.42, *p* = 0.0008) (Figure 2b). The clinico-pathological characteristics used to identify the NM-ASCC clusters were included in the multivariate Cox proportional hazards analysis, which revealed a non-independent correlation between variables, as expected. We then identified the statistically significant variables contributing to the clusters partition using a v-test based on the hypergeometric distribution to characterize the patient composition of the NM-ASCC clusters. The best prognosis Cluster 1 was characterized by female cases who received chemotherapy with a CR. While the worse prognosis clusters, Cluster 2 was composed of patients with stage III/IV disease, nodal dissemination, anal canal location, and non-CR outcome, and Cluster 3 was mainly enriched in NM-ASCC cases with HIV-HPV coinfection and non-CR outcome.

## 4. Discussion

This retrospective cohort study describes the demographic and clinical variables of NM-ASCC patients treated in two referent oncology centers using similar clinical practices for management of this disease. We analyzed a multicenter NM-ASCC cohort treated with definitive CRT and investigated clinical and pathological factors with respect to pertinent efficacy outcomes, to observe key findings regarding epidemiological features and prognostic factors of interest.

In this study, patient characteristics were well-balanced between the French and Argentinian cohorts regarding age and sex. Male sex was associated with worse OS outcomes, which is consistent with other series where significant differences according to sex in longer-term outcomes has been reported [27,28,29,30,31,32]. A notable aspect of the demographic characteristics of our population was the onset age at diagnosis. Almost half of our cohort was younger than 60 years, consistent with recently reported worldwide trends [33]. The younger age at onset may be explained by an increasing prevalence/exposure to HPV in younger populations [34]. Accordingly, more than 90% of our overall NM-ASCC cohort were HPV-positive, with a higher incidence in the French cohort that could be interpreted by geographical differences related to diagnostic methodologies applied for the HPV detection [7]. The immunohistochemistry of p16 in anal cancer, a biomarker commonly used as a surrogate for HPV involvement, and PCR to detect genotypes involved are the most widely used HPV detection methods. PCR is known to be a more sensitive and specific method, but its high cost and skill requirements are broad disadvantages [35]. In our study, the HPV status did not significantly influence CR, DFS, or OS. This could be related to the reduced number of ASCC without HPV infection detected in our overall cohort. Moreover, efforts to stratify patients according to the HPV status were made, but its place in clinical practice remains controversial given heterogeneous results for its prognostic role [36,37]. One likely possibility is that retrospective cohorts evaluated in the available meta-analysis had selection biases and confounders that influenced the results [5]. In addition, retrospective series showed discrepancies regarding the role of HIV status as a prognostic factor in ASCC [13,14]. This was presumably associated with the retrospective nature of these series considering that patients with HIV-positive tumors have typically been ineligible for the vast majority of randomized clinical trials to date [38,39]. Interestingly, our cohort had a 22% rate of HIV-positive tumors. Similar to other series [40], we did not observe differences regarding the CR rates according to HIV status, however, in contrast, we showed that HIV-positive status was a worse prognostic factor for DFS as has been reported previously [40,41,42]. Patients with HIV-HPV coinfection were also associated with a higher risk of disease progression.

Overall, 83% of the cohort had tumors arising in the anal canal. Tumor location was significantly associated with worse DFS. Very few studies distinguish between cancers of the anal canal versus perianal cancers. As such, to date it has not been possible to identify prognostic factors for a specific tumor location or subtype in the different systematic reviews. Our findings may reflect a numerical bias considering that anal canal squamous cell carcinoma is five times more common than anal margin squamous cell carcinoma. However, this result outlines the intrinsic biological differences that are poorly described to date, despite that the different ectodermic and endodermic origins of the two locations are well established.

Conflicting data exist about the role of histologic subtypes and the differentiation status, which is further complicated by the low reproducibility of identifying subtypes, and is also limited by the need for clear, well-described grading criteria. Most studies indicate that the degree of differentiation is unlikely to have an independent prognostic role in anal cancer, however, in our retrospective study, the degree of differentiation was indeed significantly associated with OS.

Of note, the overall cohort had a high number of stage III bulky tumors in contrast to several randomized clinical trials, in which the patients included for assessing CRT had an early stage of the disease [15]. In the ACT II trial, around 50% of patients had T1/T2 disease and 63% had node-negative disease [7]. Clinical practice guidelines indicate that patients could achieve a response rate of 80%, and the remaining 15% can receive abdominal perineal resection as a salvage treatment. In terms of achieving a CR, our CR rate at 6 months was 70%, which is lower than the 90.5% rate observed in the ACT II trial at 26 weeks, but is similar to prospective trials or multicentric series with similar baseline clinical staging, where the CR rates reported were between 65% and 70% for locally advanced disease [37]. We showed in our study that CR is a good surrogate of long-term outcomes as it was associated with better PFS and OS. In contrast, up to 30% of the patients who completed CRT underwent abdominal perineal resection salvage surgery. In our study, tumors greater than 5 cm and nodal involvement were significantly associated with lower rates of CR. It is not currently possible in clinical practice to predict which patients will have a favorable response to therapy after standard CRT to avoid unnecessary toxicity and comorbidities. Efforts for a deeper understanding of the biology behind this setting have seen the emergence of new biomarkers that are yet to become part of the current clinical practice guidelines.

In terms of the DFS, we confirmed that T stage (cT1-T2), N stage (cN0), HIV-negative, well-differentiated tumors, and tumor location were good prognostic factors, as supported by PFS outcomes reported in randomized trials [3,39]. Our local recurrence rate (19%) is similar to the 14.4% to 25% reported in these trials, and our distant relapse rate (7.8%) is lower than those that have been previously reported. The higher risk of distant metastasis in patients who had an incomplete response to CRT may indicate that this subgroup of patients with radiotherapy-resistant disease was a different biological subgroup, necessitating the further investigation of tumor biology [35].

In line with earlier studies, it was discovered that having a higher T stage and being male were significant predictors of worse OS [3,38,39], a higher likelihood of local failure [3,43], and locoregional failure [39]. While the greater T stage was only found to be predictive for a worse OS [38] and local failure [43], the prognostic significance of N stage disease was less apparent. According to our findings, a higher T stage is predictive of a greater likelihood of locoregional failure. Key findings are the consistent prognostic effect of the tumor size and nodal involvement stage in achieving CR, and better long-term outcomes.

In terms of the curative treatment efficacy in NM-ASCC, we observed no difference between the two referents, the geographically opposed oncology centers. Our data reflect real-world practices highlighting the need to complete CRT, with furthermore no difference between concomitant two-drug-based schedules and supervised follow-ups for locally advanced disease. In agreement with the ACTII results, our data show that performing induction followed by CRT does not positively impact long-term outcomes [19].

Our study has a number of limitations given its retrospective nature and possible selection bias for excluded cases due to missing data: different HPV methodology detections were used, and we were not able to collect detailed information regarding sexual preferences, severe adverse events, dose adjustments, or radiotherapy interruptions. While the distribution of age, sex, comorbidities, and treatment outcomes were similar between cohorts, we cannot rule out factors that bias physician decisions on the intensity of local treatment for their patients. Small numbers regarding HPV-negative tumors reinforce the need for broad collaboration in this rare disease, highlighting the importance of collecting prospective clinical and molecular data that will facilitate the validation of predictive and prognostic factors.

Due to the rarity of anal cancer, findings are frequently based on single-center studies with small cohorts, which limits the ability to uncover important prognostic markers, particularly those with a small effect size or low prevalence. Any discovered factors and their estimated effects could be biased by limited sample sizes. The predictive usefulness of well-established clinical parameters that are pertinent to current clinical practices is generally supported by this investigation. No new prognostic variables were found, but our data confirm and emphasize prognostic information considering the lack of real-world large cohorts of patients treated with CRT in multicentric studies with a prognostic factor focus.

We wanted to make sure that the prognostic factors we found were both informative of the more common HPV-driven biology and the most instructive to current therapy.

## 5. Conclusions

This international multicentric retrospective study highlights and confirms the T stage, N stage, sex, differentiation status, and tumor site as prognostic factors for outcomes after CRT in patients with NM-ASCC. Future prognostic studies can use the provided prognostic data as a starting point for variable selection. In addition, we may be able to divide patients into risk groups to design more specialized clinical trials in a translational setting by generating a collection of prognostic and maybe predictive markers for anal cancer outcomes.

## Figures and Tables

**Figure 1 biomedicines-11-00791-f001:**
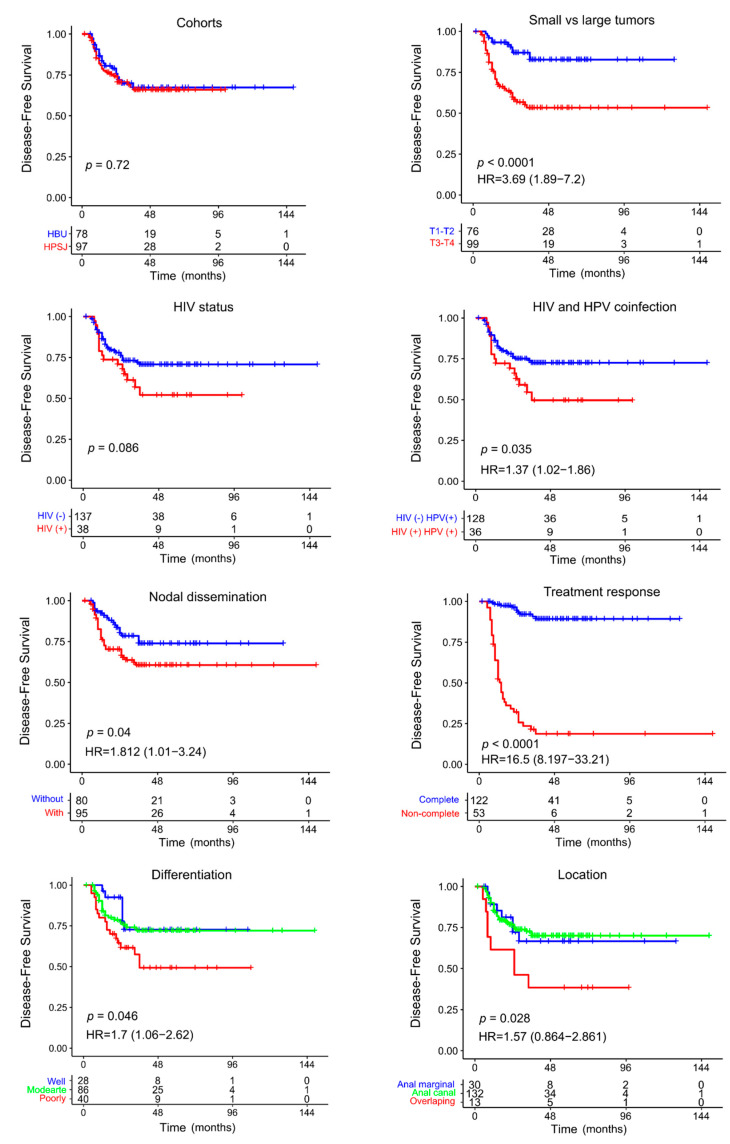
Kaplan–Meier DFS analysis and Cox regression analysis of NM-ASCC patients according to clinicopathological features.

**Figure 2 biomedicines-11-00791-f002:**
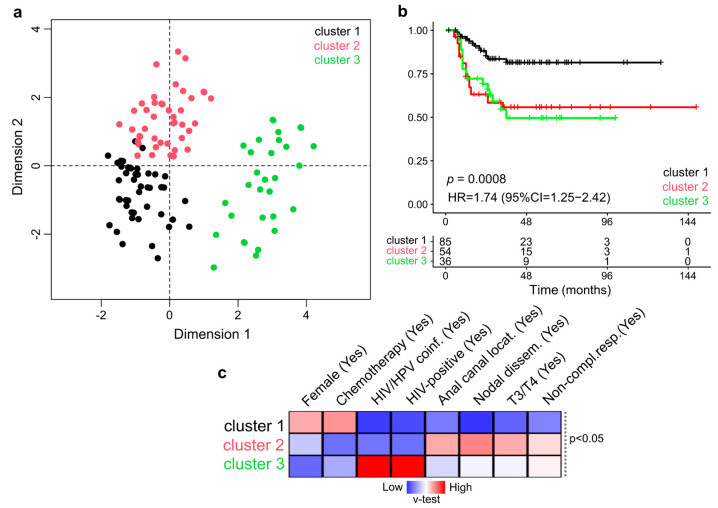
Multivariate and unsupervised analysis of clinicopathological features of NM-ASCC patients. (**a**) Multidimensional scaling plot showing the Euclidean distance of each sample from each other determined by their similarities for the included variables. The 176 patients were segregated into three classes: Cluster 1 (black), Cluster 2 (red), and Cluster 3 (green). (**b**) Univariate Kaplan-Meier survival analysis according to the assigned cluster. Survival analyses revealed that Cluster 1 was particularly associated with longer DFS compared with Clusters 2 and 3 (*p* = 0.0008). (**c**) Heatmap of the statistically significant variables (*p* < 0.05) that contributed to cluster discrimination based on positive (red) and negative (blue) v-test values.

**Table 1 biomedicines-11-00791-t001:** Clinical, demographic, treatment, and follow-up data by cohort.

Characteristics	Total *n* = 176 (%)	HPSJ *n* = 97 (%)	HBU *n* = 79 (%)	*p*-Value
Sex				0.137
Female	119 (68)	61 (63)	58 (73)	
Male	57 (32)	36 (37)	21 (27)	
Median age at diagnosis (range) in years	61 (26–89)	62 (46–89)	59 (26–87)	0.013
Location				0.006
Anal margin	30 (17)	6 (6)	24 (30)	
Anal canal	133 (76)	78 (81)	55 (70)	
Overlapping	13 (7)	13 (13)	0 (0)	
Extent of disease				0.499
cT1-T2-N0	43 (24)	20 (21)	23 (29)	
cT3-T4-N0	38 (22)	14 (14)	25 (30)	
TxcN+	95 (54)	63 (65)	32 (41)	
Differentiation				0.109
Well	29 (16)	10 (10)	19 (24)	
Medium	86 (49)	44 (45)	42 (53)	
Poorly	40 (23)	24 (25)	16 (20)	
Unknown	21 (12)	19 (20)	2 (3)	
HPV				0.006
Positive	164 (93)	95 (98)	69 (87)	
Negative	12 (7)	2 (2)	10 (13)	
HIV				0.028
Positive	38 (22)	27 (28)	11 (14)	
Negative	138 (78)	70 (72)	68 (86)	
CRT treatment response				0.689
Complete	123 (70)	69 (71)	54 (68)	
Non-complete	53 (30)	28 (29)	25 (32)	
Disease progression at follow-up				0.86
Local	37 (72)	21 (70)	16 (76)	
Distant	9 (18)	6 (20)	3 (14)	
Both	5 (10)	3 (10)	2 (10)	

HBU, Hospital Bonorino Udaondo, Buenos Aires, Argentina; HIV, human immunodeficiency virus HPSJ, Hopital Paris Saint Joseph, Paris, France; HPV, human papillomavirus. Data are presented as the number of patients (%) unless otherwise stated.

**Table 2 biomedicines-11-00791-t002:** Univariate analysis of disease-free survival (DFS) based on patient demographic, clinical and treatment characteristics.

Characteristic	Total *n* = 175 (%)	Non-Disease Progression *n* = 124 (%)	Disease Progression *n* = 51 (%)	*p*-Value
Sex				0.229
Female	118 (67)	87 (70)	31 (61)	
Male	57 (33)	37 (30)	20 (39)	
Tumor				0.0002
T1-T2	76 (43)	59 (48)	11 (22)	
T3-T4	99 (57)	65 (52)	40 (78)	
Nodes				0.035
Negative	80 (46)	63 (51)	17 (33)	
Positive	95 (54)	61 (49)	34 (67)	
Differentiation				0.037
Well	28 (16)	22 (18)	6 (12)	
Moderate	86 (49)	65 (52)	21 (41)	
Poorly	40 (23)	22 (18)	18 (35)	
Unknown	21 (12)	15 (12)	6 (12)	
HPV				0.586
Positive	164 (94)	117 (94)	47 (92)	
Negative	11 (6)	7 (6)	4 (8)	
HIV				0.047
Negative	137 (78)	102 (82)	35 (69)	
Positive	38 (22)	22 (18)	16 (31)	
HIV-HPV infection				0.018 *
HIV- and HPV-	9 (5)	5 (4)	4 (8)	
* HIV- and HPV+	128 (73)	97 (78)	31 (61)	
HIV+ and HPV-	2 (1)	2 (2)	0	
* HIV+ and HPV+	36 (21)	20 (16)	16 (31)	
CRT treatment regimen				0.001
Standard	133 (76)	103 (83)	30 (59)	
Induction + CRT	42 (24)	21 (17)	21 (41)	

Data are presented as the number of patients (%) unless otherwise stated. CRT, chemoradiotherapy. * *p*-value obtained from the comparison between HIV-HPV+ and HIV+HPV+ groups.

## Data Availability

The data presented in this study are available on request from the corresponding author. The data are not publicly available due to ethical reasons.

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
