# Peer review of "Prognostic Factors of Long-Term Outcomes after Primary Chemo-Radiotherapy in Non-Metastatic Anal Squamous Cell Carcinoma: An International Bicentric Cohort"

_biomedicines, 2023, doi:10.3390/biomedicines11030791_

Round 1

Reviewer 1 Report

Reviewer's report Manuscript ID: Biomedics 2213142 Title: Prognostic Factors of Long-Term Outcomes After Primary 2 Chemo-Radiotherapy in Non- Metastatic Anal Squamous Cell Carcinoma: An International Bicentric Cohort Date: 2023/2/6 Reviewer's report: This is an interesting manuscript as it’s a comprehensive review of the impact of CCRT affecting the prognosis as well as the long term outcomes of non-metast. anal squamous cell ca thru a multicenter study from Argentina and France. Anal squamous cell carcinoma (ASCC) stands as a rare disease, CCRT was established as the standard of care. For cT1/T2 tumors, complete regression is achieved in 80-90% of patients, with local recurrence rates of 15%, and 5-year survival of about 85% for a tumor size of ≤2 cm . In contrast, cT3-T4 and/or node-positive ASCC yields long-term recurrence rates of 20%-44% . Given the rarity of ASCC, comprehensive characterization of its molecular landscape is lacking. Thus, this study retrospectively analyzed the clinical and pathological factors affecting the local and distant DFS for non-metastatic ASCC (NM-87 ASCC) patients who have undergone curative CRT. Final conclusions reinforce the prognostic value of T stage, N stage, sex, differentiation status, tumor location, and HIV-HPV coinfection in ASCC after CCRT .I'm sure the result of this study could help in the decision-making process and guide towards an optimal therapeutic strategy for anal cancer. The MS is well prepared and containing a large amount of data. Although, there remain some limitation as this was a retrospective study with a limited number of patients. Nevertheless, it was still well written, however, a few issue need to be clarify prior publication.. 1. What was the common adverse effect encouter during CCRT of anal Reviewer's report

Manuscript ID: Biomedics 2213142
Title: 
 Prognostic Factors of Long-Term Outcomes After Primary Chemo-Radiotherapy in Non-Metastatic Anal Squamous Cell  Carcinoma: An International Bicentric Cohort 

Date: 2023/2/6

Reviewer's report:
This is an interesting manuscript as it’s a comprehensive review of the  impact of CCRT affecting the prognosis as well as the long term outcomes of non-metast. anal squamous cell ca thru a multicenter study from Argentina and France. 
Anal squamous cell carcinoma (ASCC) stands as a rare disease, CCRT was established as the standard of care. For cT1/T2 tumors, complete regression is achieved in 80-90% of patients, with local recurrence rates of 15%, and 5-year survival of about 85% for a tumor size of ≤2 cm . In contrast, cT3-T4 and/or node-positive ASCC yields long-term recurrence rates of 20%-44% . Given the rarity of ASCC, comprehensive characterization of its molecular landscape  is lacking. Thus, this study retrospectively analyzed the clinical and  pathological factors affecting the local and distant DFS for non-metastatic ASCC (NM-87 ASCC)patients who have undergone curative CRT.  Final conclusions  reinforce the prognostic value of T stage, N stage, sex, differentiation status, tumor location, and HIV-HPV coinfection in  ASCC after CCRT .I'm sure the result of this study could help inthe decision-making process and guide towards an optimal therapeutic strategy for anal cancer.

The MS is well prepared and containing a large amount of data.  Although, there remain some limitation as this was a retrospective study with a limited number of patients.  Nevertheless, it was still well written, however, a few issue need to be clarify prior publication..

1. What was the common adverse effect encouter during CCRT of anal ca.? and 

 what was the severity of the adverse effect ?

Author Response

We thank the reviewer for their effort in reviewing our manuscript and considering our study as relevant from a scientific and clinical point of view. We understand the reviewer's concern regarding the importance of toxicity-related treatment. Unfortunately, due to the study's retrospective nature, it was not designed to collect the information properly in both centers homogeneously in the medical records. Therefore, the data collected regarding adverse events would not be accurate to report, we have explicitly stated this as a study limitation in the manuscript.

Reviewer 2 Report

 This study assessed prognostic factors after chemoradiotherapy for non-metastatic anal squamous cell carcinoma. The inquiry is significant due to the rarity of anal squamous cell carcinoma.

Major Points:

1.       Kindly furnish the results differentiated by chemotherapy regimen. Given that the accessibility of MMC has been challenging, it would be insightful to juxtapose the outcomes of 5-FU + MMC, which constituted the standard of care, with those of other regimens.

2.       Please elaborate on the radiotherapy regimen. Do variations in the dose delivered yield differing treatment outcomes?

3.       In view of the fact that the majority of cases in this study were HPV-positive, the results with and without HPV-HIV co-infection and with and without HIV positivity were largely similar (Fig. 1, Table 2). The authors declare that HIV-HPV co-infection constitutes a risk. To conclusively affirm this, it might be imperative to contrast the following four groups: HPV(+)-HIV(+), HPV(+)-HIV(-), HPV(-)-HIV(+), HPV(-)-HIV(-).

4.       The figures for HIV-HPV co-infection and non-co-infection in Table 2 appear to be inverted.

5.       Could you kindly verify that the numbers in Figure 1 and Table 2 are accurate? While Table 2 reports 164 HPV-positive and 12 negative, Figure 1 reports 139 HPV(+)-HIV(-) and 36 HPV(+)-HIV(+) – equating to 175 HPV-positive cases.

6.       The "Location" in Figure 1 appears to demonstrate that the group that overlaps has subpar performance. Is this not analogous to saying that those with substantial T-stages have inferior performance?

7.       With reference to Figure 2, it is stated in the text that no significant disparity was discovered in the CR rate, DFS, and OS between the HPSJ and HBU cohorts. Why was the Hospital included as a stratification factor?

8.       Would it not be more appropriate not to incorporate treatment facilities as a stratification factor for the clusters, thus rendering the results more widely applicable?

Minor Points:

1. A half-width space should be inserted immediately following the inequality sign.

2. In the Introduction section, there is an ambiguity between the term T1/T2 and the term T3-T4. It is preferable to maintain consistency.

3. In the Result section, the abbreviation "NM" suddenly appears, presumably referring to Non-metastatic, but it is more expedient to spell it out upon its initial appearance in the text.

Author Response

We thank the reviewer for their effort in reviewing our manuscript and your valuable feedback. we believe your questions and suggestions will bring major improvements to our article

Major Points:

  1. Reviewer:Kindly furnish the results differentiated by chemotherapy regimen. Given that the accessibility of MMC has been challenging, it would be insightful to juxtapose the outcomes of 5-FU + MMC, which constituted the standard of care, with those of other regimens.

Author response: As we referred in the paragraph treatment and follow-up, all patients have received a two chemotherapy-based scheme concurrently to radiotherapy. The results showed that 88% of patients had received Mitomycin C with 5FU (65%) or Capecitabine (23%). As the reviewer has mentioned MMC and 5FU have been the standard of care. Anyway, several publications have demonstrated no inferiority between selecting 5 FU or Capecitabine associated with MMC in the CRT approach for ASCC. Thus both schemes based on MMC associated with 5FU or its analog Capecitabine are widely recommended as an initial therapeutic approach. Only 12% have received Cisplatin with 5FU instead of Mitomycin. This is an alternative recommended according to the risk of higher myelosuppression associated with MMC in selected patients i.e., HIV associated with a low CD4 count. Due to the power limitation to compare the different schemes due to the different sample numbers, we haven't performed a comparison between schemes of CRT.

In contrast, when we described the patient who had received induction chemotherapy before CRT, as we have stated, it was also a physician discretion criteria based on bulky disease, rapid relief of symptoms or comorbidities. Once again, the numbers are not well balanced according to the chemotherapy scheme selection due to the retrospective nature of the study. But we have detailed all the treatments because it represents the daily clinical practice of a real-world cohort. In table 2, we described a significant p-value (p=0.001) between standard CRT and induction + standard CRT. In the discussion, we also described that induction followed by CRT results are similar to the previous series without positively impacting long-term outcomes.

We understand that daily practice could be a bias. That's why we have initially compared the oncological outcomes between both cohorts without observing statistical differences. After obtaining this statement, we assumed both cohorts could be analyzed focusing on prognostic factors for oncological outcomes.

  1. Reviewer:Please elaborate on the radiotherapy regimen. Do variations in the dose delivered yield differing treatment outcomes?

Author response: Unfortunately, we cannot provide an accurate analysis of RT regimen. All the planning RT data is not available for all patients to review retrospectively. The information regarding the initial plan dose, interruptions, or toxicities associates was not homogeneously recorded across the cases. We have mentioned in the study limitations section the lacking of this information. Due to its retrospective nature, RT planning was assigned case by case according to the initial cTNM. According to our eligibility criteria, we have selected all patients that have completed the treatment to avoid other biases regarding the CRT and described the median dose of RT delivered.

Besides, knowing the design of the study and potential bias regarding schemes and toxicity, was not our scope to describe efficacy differences regarding treatments. But we have ensured at first that the outcomes between both cohorts were similar, as stated in the manuscript and Figure 1.

  1. Reviewer:In view of the fact that the majority of cases in this study were HPV-positive, the results with and without HPV-HIV co-infection and with and without HIV positivity were largely similar (Fig. 1, Table 2). The authors declare that HIV-HPV co-infection constitutes a risk. To conclusively affirm this, it might be imperative to contrast the following four groups: HPV(+)-HIV(+), HPV(+)-HIV(-), HPV(-)-HIV(+), HPV(-)-HIV(-). 

Author response: As suggested by the reviewer, the four groups derived from HIV and HPV combinations were introduced in Table 2 and Supplementary table 2. However, prevalence and follow-up analyses were done between the two representative groups: HIV(-)-HPV(+) vs. HIV(+)-HPV(+) due to sample size limitations in the HPV-negative groups. HPV infection is a relevant risk factor associated with ASCC development, and for that HPV-negative subgroup is usually represented by a small number of patients across all series.

  1. Reviewer:  The figures for HIV-HPV co-infection and non-co-infection in Table 2 appear to be inverted.

Author response: We thank the reviewer for their detailed observations, Table 1, 2 and supplementary tables were carefully revised. 

  1. Reviewer:  Could you kindly verify that the numbers in Figure 1 and Table 2 are accurate? While Table 2 reports 164 HPV-positive and 12 negative, Figure 1 reports 139 HPV(+)-HIV(-) and 36 HPV(+)-HIV(+) – equating to 175 HPV-positive cases.

Author response: Thanks for your comments. All data in the new manuscript version has been duly revised. 

  1. Reviewer: The "Location" in Figure 1 appears to demonstrate that the group that overlaps has subpar performance. Is this not analogous to saying that those with substantial T-stages have inferior performance?

Author response: The location was related to the distance to the anal verge in each case  moreover the cTNM. So it would not be analogous for us.

7 and 8. Reviewer:  With reference to Figure 2, it is stated in the text that no significant disparity was discovered in the CR rate, DFS, and OS between the HPSJ and HBU cohorts. Why was the Hospital included as a stratification factor?

Would it not be more appropriate not to incorporate treatment facilities as a stratification factor for the clusters, thus rendering the results more widely applicable?

Author response: We thank the reviewer’s advice regarding excluding the treatment facilities in the multivariate model. We have re-done the HCPC analysis without the treatment facilities and introduced it in the manuscript's revised version, arriving at the same results to be more widely applicable. Moreover, the outcomes were not different between cohorts, perhaps keeping the facilities would have been rhetoric because the main results achieved in the model are the prognostic factors that intrinsically are represented in both centers.

Minor Points:

  1. Reviewer: A half-width space should be inserted immediately following the inequality sign. 

Author response: The suggestion was introduced in the manuscript.

  1. Reviewer: In the Introduction section, there is an ambiguity between the term T1/T2 and the term T3-T4. It is preferable to maintain consistency. 

Author response: This suggestion was accepted and we refer as T1-T2 and T3-T4 in the introduction in the new manuscript attached. 

  1. Reviewer: In the Result section, the abbreviation "NM" suddenly appears, presumably referring to Non-metastatic, but it is more expedient to spell it out upon its initial appearance in the text.

Author response: In the introduction, in the sentence 87, at the end of the last paragraph, NMASCC is defined. “We performed a retrospective multicenter study aimed at elucidating clinical and pathological factors impacting local and distant DFS for non-metastatic ASCC (NM-ASCC) patients who have undergone curative CRT”

Reviewer 3 Report

While this study doesn't find anything particularly new, it does contribute additional patient data for a relatively rare disease site and points to some interesting avenues of future research, so I think it's worth publishing despite the limitations, which the authors cover well in the discussion. My main concern is with the presentation of the results and making sure all of the results in the tables and figures are actually correct. In Table 1, the HIV Negative percentage for HPSJ is displayed as 2 when it should actually be 82. In the last row of this table, Disease progression at follow-up, there are only 3 options listed (Local, Distant, Both) yet there are 4 rows of numbers for the 3 options, and also the percentages here do not add up to 100%. There are probably other errors in this data as well, as there are other columns where the percentages do not add up to 100%. For better clarity, the percentage values should actually have a percent sign (%) after them. In Table 2, I disagree with how the percentages are calculated. In Table 1, the purpose was to compare two cohorts to make sure they were the same across different variables, so it made sense to have the percentages calculated one way, so that the columns add to 100% instead of the rows. But for Table 2, the percentages should be calculated across the columns, the way they are calculated in the Supplementary tables. In fact the final entries in Table 2 (the CRT treatment regimen) are calculated this way. All of Table 2 should have the percentages calculated as they are in the CRT treatment regimen section. Again, % signs would help with clarity. Figure 1 seems fine except for the figure on Differentiation. I was very surprised to look at this figure and see the Kaplan-Meier curve for Well-differentiated showed lower DFS than Moderate and Poorly-differentiated. I see this in Supplementary figure 1 as well. But in the results and discussions, the authors say that Well-differentiation is a positive prognostic factor. I assume the figures must be wrong, perhaps the labels were switched for Well and Poorly differentiated? This must be confirmed by the authors and fixed. Other than these issues with the tables and figures, the paper is well-written with the limitations and strengths clearly noted, so I don't have any comments on the text.

Author Response

We deeply appreciate all your comments and suggestions. We have carefully revised Table 1, Table 2, Supplementary tables and Figure 1 in the new manuscript version. We have considered all your suggestions and fixed the errors mentioned. Please find enclosed all the changes clarified In the new manuscript version uploaded.

Round 2

Reviewer 2 Report

Thank you for your attentive response to my suggestions and inquiries. I am confident that this revised manuscript has been sufficiently improved to warrant publication.

Author Response

Thanks for your interest in our paper.  All sentences reported as high similarity were edited at introduction, results, discussion and conclusion. Also in the limitations paragraph we have added the different HPV methology detection as was suggested by the editor.